# Variability Study of Bond Work Index and Grindability Index on Various Critical Metal Ores

**Gloria G. García [1], Josep Oliva [2], Eduard Guasch [2], Hernán Anticoi [3], Alfredo L. Coello-Velázquez [4] and Juan M. Menéndez-Aguado [1,\*]**

[1] Escuela Politécnica de Mieres, University of Oviedo, Gonzalo Gutiérrez Quirós, 33600 Mieres, Spain; gloria.glez.gcia@gmail.com or UO150088@uniovi.es

[2] Departament D'Enginyeria Minera, Industrial i TIC, Universitat Politècnica De Catalunya Barcelona Tech, Av. Bases De Manresa, 08242 Manresa, Spain; josep.oliva@upc.edu (J.O.); eduard.guasch@upc.edu (E.G.)

[3] Escuela Politecnica de Ingenieria de Minas y Energia, Universidad de Cantabria, Ronda Rufino Peón, 39316 Torrrelavega, Spain; hernan.anticoi@unican.es

[4] CETAM, Universidad de Moa Dr. Antonio Núñez Jiménez, Moa 83300, Cuba; acoello@ismm.edu.cu

\* Correspondence: maguado@uniovi.es; Tel.: +34-985458033

**Abstract:** It is a well-known fact that the value of the Bond work index ($w_i$) for a given ore varies along with the grinding size. In this study, a variability bysis is carried out with the Bond standard grindability tests on different critical metal ores (W, Ta), ranging from coarse grinding (rod mills) to fine grinding (ball mills). The relationship between $w_i$ and grinding size did not show a clear correlation, while the grindability index ($gpr$) and the grinding size showed a robust correlation, fitting in all cases to a quadratic curve with a very high regression coefficient. This result suggests that, when performing correlation studies among ore grindability and rock mechanics parameters, it is advised to use the grindability index instead of the Bond work index.

**Keywords:** grindability; comminution; Bond work index

## 1. Introduction

Comminution is an essential operation for the mining and mineral processing industry. It also plays a central role in the cement production, ceramics and chemical industries. In the mineral industry, the liberation of valuable minerals from the gangue is a fundamental requirement for all subsequent separation or extraction operations, and this is achieved through several stages of rock fragmentation, that is, by comminution of the ore [1].

Schönert [2] estimated that minerals comminution consumes 3% of all the energy produced by industrialized countries, in line with former studies [3]. More recent evaluations estimate that comminution operations are responsible for 3–5% of energy consumption at a global scale [4]. Moreover, in terms of OPEX in mineral processing plants, comminution operations amount to 40–50% of the energy consumption.

Considering the above, any gain in efficiency can significantly impact the plant operating costs and the consequent conservation of resources [5]. In this sense, an improvement in knowledge of ore grinding behavior can allow modification of the operation and control strategies of the grinding operations, resulting in significant energy savings. This would increase the competitiveness of operations and reduce emissions.

It is common to process multicomponent ores, made up of at least two mineralogical components with differences in their physical and physicomechanical properties. Some authors [6,7] show that disregarding the variability of the feed mineralogical composition produces large deviations in the planned metallurgical efficiency, along with problems in the treatment of the ores with such characteristics. On the other hand, between the initial

exploration work for the design of any mineral beneficiation plant and the reaching of its full operating regime, and even after reaching it, there will be variations in the plant feed composition, implying substantial changes in mineral properties. Therefore, it would be advisable to adjust the operating and control conditions of the treatment plant in general and the size reduction section in particular.

The energy–size relationships in comminution processes have been an object of research since the first industrial revolution [8]. Rittinger [9] proposed the first law of comminution, supposing that the amount of created surface is proportional to the specific energy consumption in grinding operations, as expressed in Equation (1):

$$E = K_R \cdot \left( \frac{1}{P} - \frac{1}{F} \right)$$

(1)

where $E$ is the specific energy consumption [kWh/t], $K_R$ is the proportionality coefficient and $P$ and $F$ are the particle sizes of the product and feed, respectively [$\mu$m].

Kick [10], in the second law of comminution, argued that according to his calculations, the specific energy consumption would be proportional to the volume reduction, as expressed in Equation (2), where $K_K$ is a different proportionality coefficient.

$$E = K_K \cdot \left( \frac{1}{\ln(P)} - \frac{1}{\ln(F)} \right)$$

(2)

The differences between the Rittinger and Kick models lasted for years, until the proposal of the third theory of comminution by F. Bond [11–13], which is summarized in Equation (3),

$$E = K_B \cdot \left( \frac{1}{\sqrt{P}} - \frac{1}{\sqrt{F}} \right)$$

(3)

where $K_B = 10 \cdot w_i$, and $w_i$ is expressed in kWh/t.

Subsequent studies [14] explained that the three laws derive from a generalized comminution differential equation, each one best applied to a different size range (Rittinger's law for fine grinding; Bond's law for coarse grinding and secondary/tertiary crushing; and Kick's law for primary crushing). The novelty in the third law's proposal was the procedure for determining $w_i$ in the case of crushing, rod milling and ball milling [13,15]. The practical interest of $w_i$ is unquestionable. From a technical perspective, it constitutes the most reliable method of characterizing ore grindability when designing the necessary tumbling mills to process that ore. Bearman et al. (1997) showed that other mechanical characterization tests are insufficient when predicting the grinding ore behavior.

A logical reasoning process should suggest finding some correlation among mechanical parameters (hardness, Young's modulus, uniaxial compression strength (UCS), etc.) and the ore grinding behavior. Several researchers [4,16–18] followed that inspiration, but no generalizable results have been obtained since grindability behavior is usually evaluated under closed-circuit conditions, which means that not only breakage but breakage plus classification operations are involved. Moreover, we can easily find ores with high hardness and high grindability values, but among the highest grindability values, we can find quite soft ores (graphite or mica group minerals). On the other hand, diamond mineral shows modest grindability values. Thus, it is worth emphasizing that the Bond work index tries to characterize the ore grinding behavior in a closed circuit, encompassing the ore mechanical behavior before the mill action (i.e., whatever the type of the mill and its characteristics of action), but also the screening or classification stage involved in the closed circuit, which is greatly influenced by product size and shape.

Due to the fact that Bond's proposal was undoubtedly linked to a market-dominant firm, i.e., Allis Chalmers, which even owned the patent of the standard mill, several proposals soon emerged to define alternative test approaches, which can be grouped in the following types:

- Indirect work index determination in other lab mills [19–24].
- Specific energy determination from correlations in different devices [25–27].
- Work index calculation through lab tests and simulation [28–31].

It is essential to notice that despite the almost unanimous consideration of the $w_i$ as the characteristic parameter of ore grinding behavior, it is not fully understood at the industrial level, even being handled as a constant value. Bond himself usually reported in his papers separately the grindability values for the Bond rod mill test (BRM) and the Bond ball mill test (BBM), but no study could be found analyzing the information from BRM and BBM test values and deepening them to explain the variability obtained.

In this work, the analysis of grindability results obtained in a broad particle size range and several critical metal ores (W, Ta) is carried out. The variability of the work indices in BRM and BBM tests is studied to propose a methodology to model said variability.

## 2. Materials and Methods

### 2.1. Materials

This study was carried out on three ores from W mines and two ores from a Ta mine. Two of the W ore samples were Scheelite ores, received from Barruecopardo (Spain) and Mittersill (Austria). A detailed description of Barruecopardo ore can be found in recent publications [32,33]. In the case of Mittersill, an ore description can be found in [34]. The third W ore was Wolframite from the Panasqueira Mine (Portugal), and a detailed description of this ore can be found in [35]. In the case of Ta ores, two different samples were received from Penouta Mine (Spain), one from the open pit and the other one from the tailings pond of the former Tin mining activities in that mine. Characterization studies of those samples have been previously published [36,37]. It must be pointed out that, in the particular case of Barruecopardo mine, two different samples were taken from different heaps. The sample size in each case, considering the largest particle size, was enough to perform the series of Bond ball mill grindability tests separately (see Section 2.2), but not enough to perform the series of Bond rod mill grindability tests separately for each sample (see Section 2.3). Accordingly, it was decided to blend and homogenize the Barruecopardo samples and perform the rod mill test on the samples blend.

### 2.2. Bond Ball Mill (BBM) Standard Test

The procedures to carry out the Bond grindability tests in ball mills and rod mills are outlined in Sections 2.2 and 2.3. They are usually referred to as the standard tests, but it must be highlighted that the procedures haven't been defined by ISO or ASTM standards. The closest attempt to a standard definition was the initiative of the Global Mining Standard Group [38].

The Bond work index most commonly referred to is the BBM work index. This value is obtained in a 12″ × 12″ laboratory mill running at 70 rpm, with rounded inner edges and without lifters. The grinding charge is comprised of a distribution of steel balls with several diameters. Table 1 shows the original Bond proposal [13], while the last Bond recommendation can be found in Table 2 [39].

**Table 1.** Ball grinding charge distribution proposed by Bond.

| Ball Size | | Balls | |
|---|---|---|---|
| Inch | cm | Number | Weight (g) |
| 1.45 | 3.683 | 43 | 8803 |
| 1.17 | 2.972 | 67 | 7206 |
| 1.00 | 2.540 | 10 | 672 |
| 0.75 | 1.905 | 71 | 2011 |
| 0.61 | 1.549 | 94 | 1433 |
| | Total: | 285 | 20,125 |

**Table 2.** Ball charge distribution used in this research.

| Ball Size | | Balls | |
|---|---|---|---|
| Inch | cm | Number | Weight (g) |
| 1.500 | 3.810 | 25 | 5690 |
| 1.25 | 3.175 | 39 | 5137 |
| 1.000 | 2.540 | 60 | 4046 |
| 0.875 | 2.223 | 68 | 3072 |
| 0.750 | 1.905 | 93 | 2646 |
| | Total: | 285 | 20,592 |

The mill feed must be prepared by controlled crushing until 100% passes through a 6 Tyler mesh (3.35 mm). The first grinding cycle feed must be 700 cm³, and this volume's weight is fixed as the mill charge in all subsequent cycles. Additionally, fresh feed particle size distribution (PSD) is obtained to calculate the 80% passing size ($F_{80}$) and undersize weight already present in the feed.

The test procedure consists of performing several dry grinding cycles to simulate a continuous closed-circuit operation with 250% circulating load (Figure 1). The circuit is closed by a sieve ($P_{100}$) selected according to the industrial grinding size target, always between 28 and 325 Tyler mesh (40–600 microns).

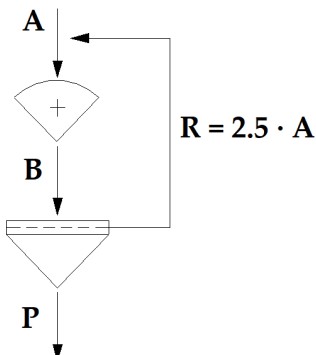

**Figure 1.** Closed-circuit BBM test objective layout.

The first cycle starts with an arbitrary number of mill revolutions, usually 100 revolutions with hard-to-grind ores and 50 revolutions with soft ores. The first run product is sieved, the undersize is weighed, and the net grams produced (*gpr*) of the first run is calculated, considering the undersize already present in the feed.

The second cycle feed is constituted by the former cycle's oversize product plus enough fresh feed to complete the initial 700 cm³ weight. The second cycle number of revolutions is calculated considering the predefined circulating load value (250%), according to Equation (4),

$$n_i = \frac{(P_S - F_{f,i})}{gpr_{i-1}} \tag{4}$$

where $n_i$ is the number of mill revolutions at run $i$; $P_S$ is the expected product weight once it reaches the steady state (g), calculated by dividing the initial 700 cm³ weight by 3.5; $F_{f,i}$ is the weight of fines already in the feed (g), which can be calculated from the feed PSD and the total fresh feed weight added in the run $i$ (which equals the total undersize product in the run $i−1$) and $gpr_{i-1}$ is the net grams produced in the previous run, $i−1$.

Subsequent grinding cycles are carried out (at least five) until $gpr$ reaches equilibrium. The final value of $gpr$ is calculated as the average of the last three cycles. The final cycle product PSD is calculated to obtain $P_{80}$, and the BBM work index can be calculated using Equation (5),

$$w_i = \frac{44.5}{P_{100}^{0.23} \cdot gpr^{0.82} \cdot \left( \dfrac{10}{\sqrt{P_{80}}} - \dfrac{10}{\sqrt{F_{80}}} \right)} \tag{5}$$

where the BBM work index, $w_i$, is expressed in kWh/sht; $P_{100}$, $F_{80}$ and $P_{80}$ are expressed in microns and $gpr$ is expressed in g/rev. Bond named $gpr$ as the grindability index.

According to Bond [13], $w_i$ should conform with the motor output power to an average overflow ball mill of 8 ft inner diameter grinding wet in a closed circuit. This value should be multiplied by correcting factors to conform with other situations, such as dry grinding (at least 1.30) or different inner mill diameters. A complete and updated description of correction factors was written by Rowland [40].

### 2.3. Bond Rod Mill (BRM) Standard Test

In this case, the procedure is very similar to BBM test, and only some differences are commented on [13]. The feed must be prepared until 100% passes ½″ (1, 27 mm), with a feed volume of 1250 cm³. Dry grinding cycles are performed with 100% circulating load in a laboratory rod mill 12″ × 24″ with a wave-type lining, running at 46 rpm. The grinding charge consists of six 1.25″ diameter and two 1.75″ diameter steel rods 21″ long, weighing 33.380 kg. In this case, $P_{100}$ values can range from 4 to 65 Tyler mesh (4.7 mm to 200 microns).

In order to equalize segregation at the mill ends, it is rotated level for eight revolutions, then tilted up 5° for one revolution, down 5° for another revolution and returned to level for eight revolutions continuously through each grinding cycle. At the end of each cycle, the mill is discharged by tilting downward at 45° for 30 revolutions. Once equilibrium is reached, $gpr$ and $P_{80}$ are calculated, and the BRM work index is calculated from Equation (6).

$$w_i = \frac{62}{P_{100}^{0.23} \cdot gpr^{0.625} \cdot \left( \dfrac{10}{\sqrt{P_{80}}} - \dfrac{10}{\sqrt{F_{80}}} \right)} \tag{6}$$

Again, $w_i$ should conform with the motor output power to an average overflow rod mill of 8 ft inner diameter grinding wet in an open circuit.

### 2.4. Grindability Tests

A series of tests was defined to analyze the variation of grindability properties in the selected ores. Depending on sample availability, a minimum of three BBM tests and a minimum of 2 BRM tests were performed, each test at a different $P_{100}$ for every ore. Then,

the values of *gbp* and $w_i$ were obtained for each ore, and an attempt to model their variation with $P_{100}$ was performed in each case. Full details of the performed tests and results are available in the Supplementary Materials.

It is generally accepted that, provided samples are representative, BBM and BRM grindability test repetitions are unnecessary. This is justified by the iterative nature of the grindability tests procedures, and both rod and ball mill tests' repeatability were proven to be less than ±4% at two standard deviations [41].

### 3. Results and Discussion

In the case of Penouta tailings pond ore, the variation of $w_i$ versus $P_{100}$ is plotted in Figure 2, for both BBM and BRM tests. The obtained values show a lack of continuity, and a clear trend function could hardly be defined. Nonetheless, when observing Figure 3, which depicts the variation of *gpr* versus $P_{100}$ in both BBM and BRM tests, a fairly clear trend can be seen; according to this, Figure 3 also shows the quadratic fit of *gpr* consolidated values versus $P_{100}$, with a determination coefficient of 99.76%.

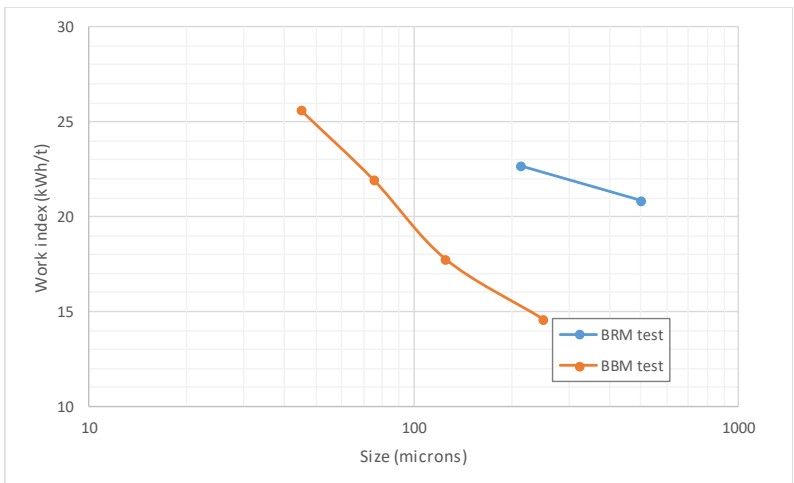

**Figure 2.** Variation of BBM and BRM $w_i$ values with $P_{100}$, Penouta tailings pond ore.

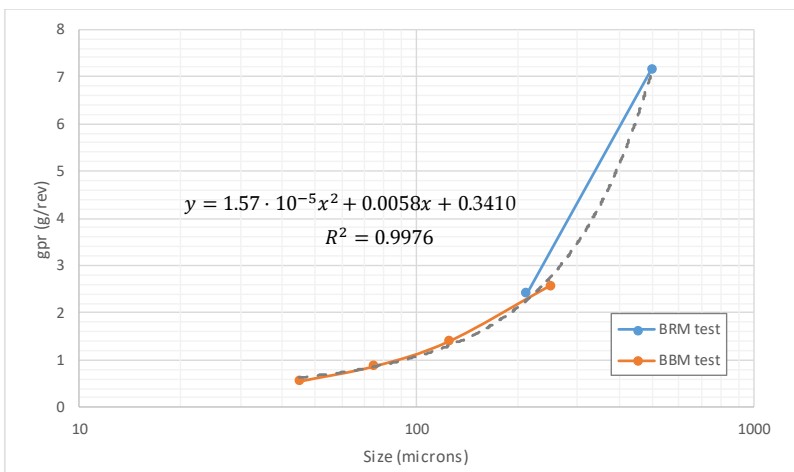

**Figure 3.** Variation of BBM and BRM *gpr* values with $P_{100}$, Penouta tailings pond ore.

A similar analysis was performed in the case of Penouta mine ore (see Figures 4 and 5). In this case, despite $w_i$ versus $P_{100}$ plot revealing a lack of continuity again (Figure 4), plotting *gpr* versus $P_{100}$ (Figure 5) showed a similar trend to the previous ore. Moreover,

the quadratic fit was almost perfect in this case, with a coefficient of determination of 100.00%.

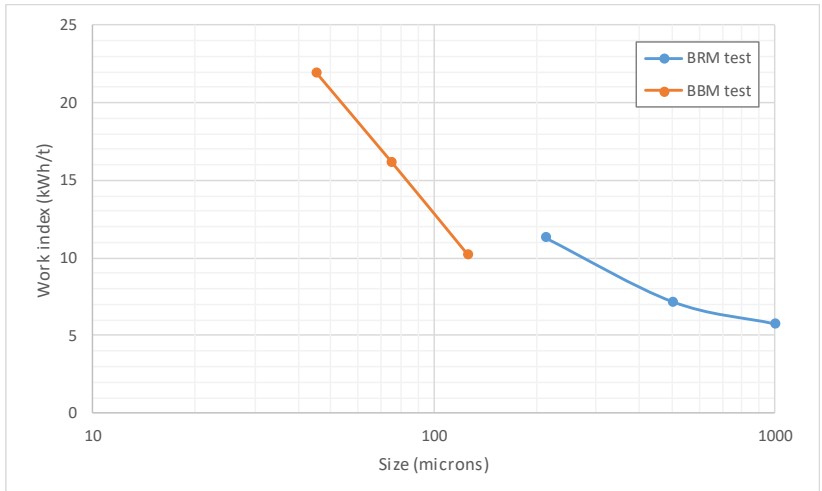

**Figure 4.** Variation of BBM and BRM $w_i$ values with $P_{100}$, Penouta mine ore.

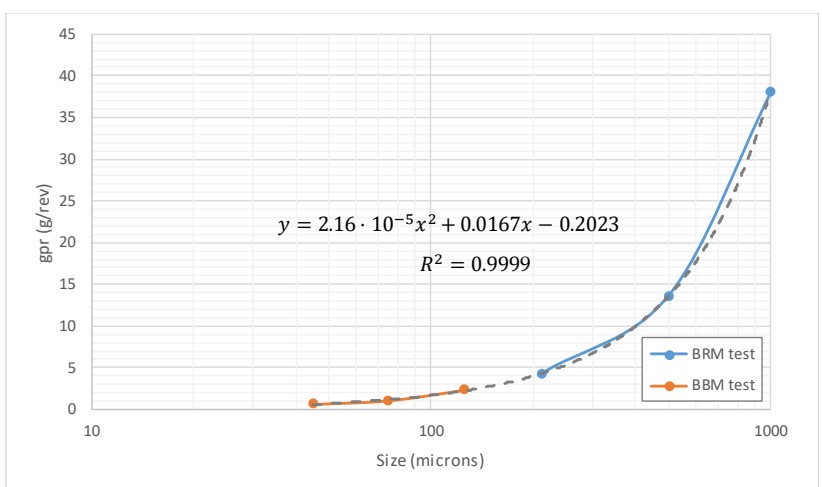

**Figure 5.** Variation of BBM and BRM $gpr$ values with $P_{100}$, Penouta mine ore.

In the case of Mittersill ore (Figures 6 and 7), the transition between BBM and BRM $w_i$ values with $P_{100}$ shows a better continuity than in previous cases (Figure 7), so the determination coefficient reached again a very high value, 99.89%.

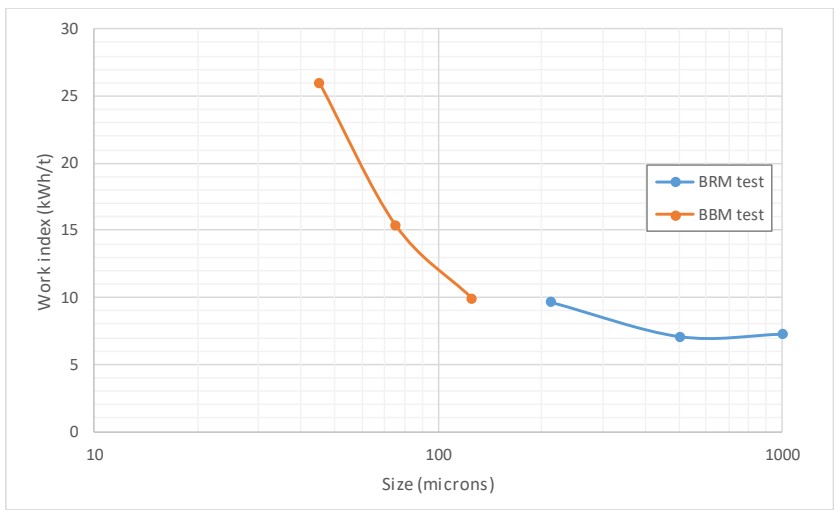

**Figure 6.** Variation of BBM and BRM $w_i$ values with $P_{100}$, Mittersill ore.

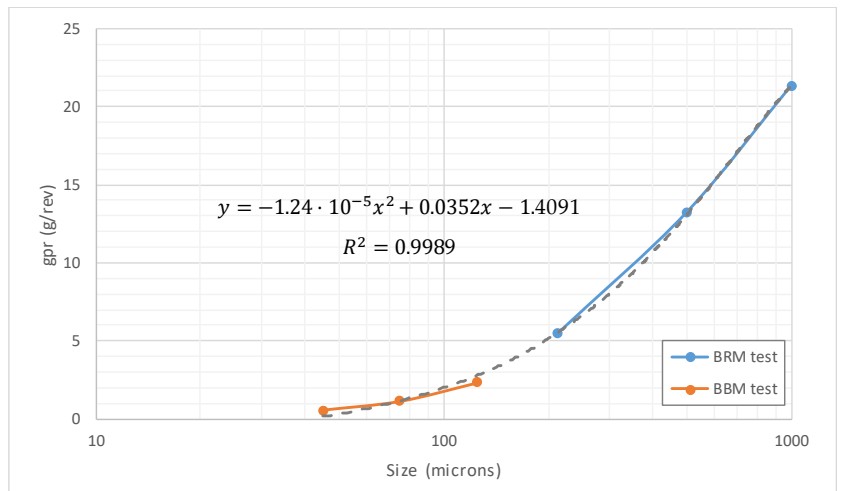

**Figure 7.** Variation of BBM and BRM $gpr$ values with $P_{100}$, Mittersill ore.

Plotting BBM and BRM $w_i$ values versus $P_{100}$ in the case of Panasqueira ore yeilded a clear trend in the case of BBM $w_i$ values, but a with a roller-coaster type shape in the case of BRM $w_i$ values (Figure 8). Unexpectedly, when plotting $gpr$ values versus $P_{100}$ (Figure 9), again a quadratic fit yielded a very high value of the determination coefficient, 99.10%.

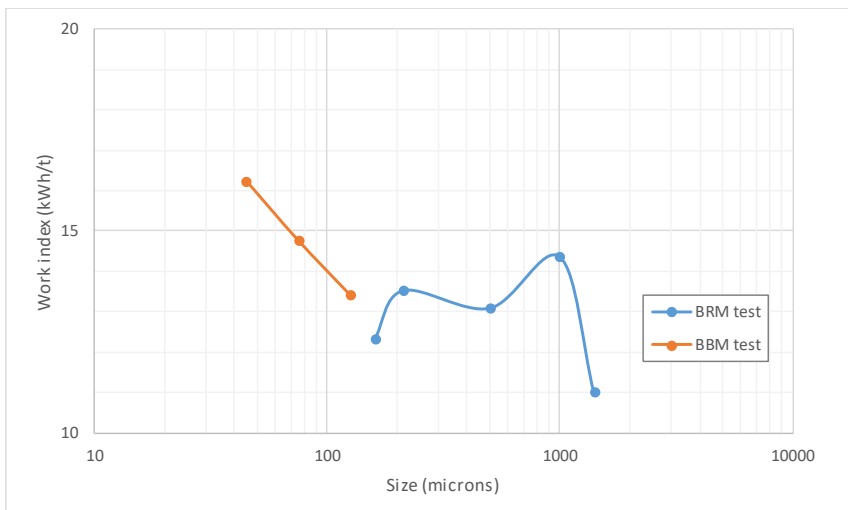

**Figure 8.** Variation of BBM and BRM $w_i$ values with $P_{100}$, Panasqueira ore.

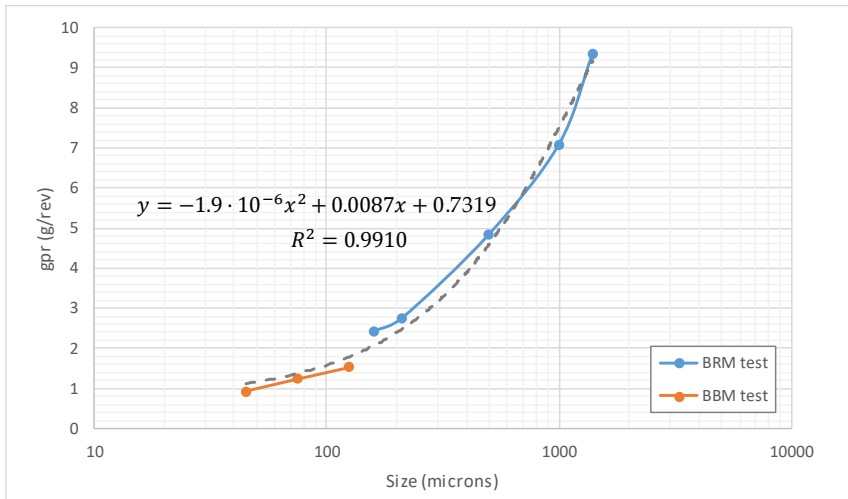

**Figure 9.** Variation of BBM and BRM $gpr$ values with $P_{100}$, Panasqueira ore.

Finally, Barruecopardo ore samples results are depicted in Figures 10 and 11. As mentioned above, BBM tests were performed on the same ore samples but with different origins, while BRM tests were performed on the composite obtained after blending both samples. Once more, with an evident lack of continuity in the $w_i$ versus $P_{100}$ plot (Figure 10), a clear quadratic trend was obtained when plotting $gpr$ values versus $P_{100}$ (Figure 11), with a very high value of the determination coefficient, 99.95%.

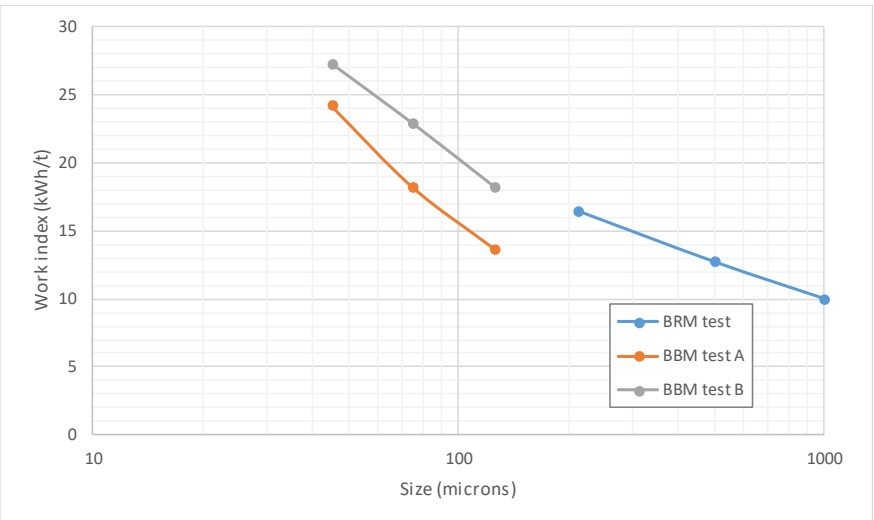

**Figure 10.** Variation of BBM and BRM $w_i$ values with $P_{100}$, Barruecopardo ore.

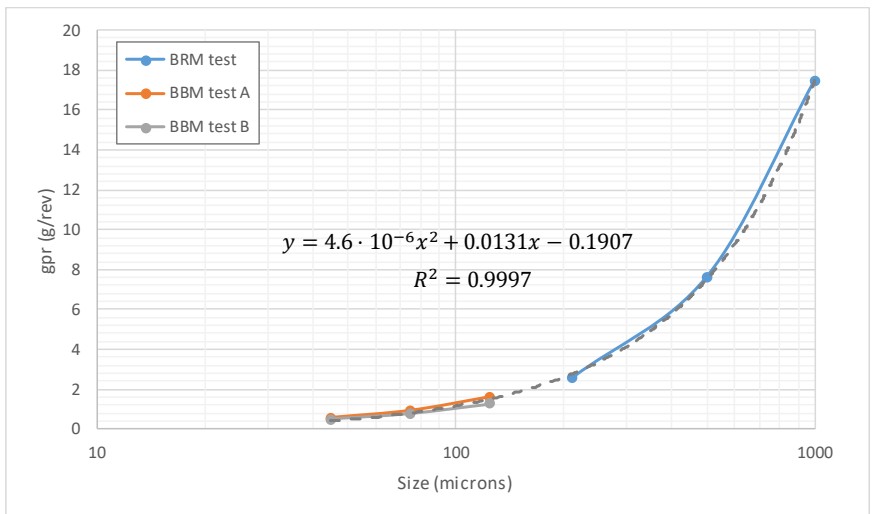

**Figure 11.** Variation of BBM and BRM $gpr$ values with $P_{100}$, Barruecopardo ore.

Given the results obtained, it is evident that there is significant variability of $w_i$ values with grinding size, both in BBM and BRM grindability tests. While $w_i$ versus $P_{100}$ plots show no continuity in general (being erratic in the case of Panasqueira ore, BRM $w_i$ values) when plotting $gpr$ versus $P_{100}$, a parabolic shape is clearly depicted with all ores. Furthermore, the quadratic fitting determination coefficients overcame 99.7% in all cases.

A recommendation can be made in the light of these results: any energy consumption model based on correlating $w_i$ with mechanical parameters (geotechnical) or operational parameters (drilling, blasting) should be revised considering $gpr$ values instead of $w_i$ values, which probably would yield a better determination coefficient.

These results also invited us to perform a conceptual review of the Fred Bond literature to seek relevant considerations about the significance of $gpr$. Thus, it gains additional value that $gpr$ was already named "grindability" since the paper led by Walter Maxson [42], in which Fred Bond was also a co-author. Fred Bond, in his subsequent papers, also utilized this definition. Considering that $w_i$ is worldwide known as the Bond index, and without the intention of subtracting an iota of importance from the broad contribution of Fred Bond ($w_i$ is the most practical tool in rod and ball mill calculation), it seems fair to propose the naming of $gpr$ as the *Maxson index*. This so-called Maxson index should be

meaningful, not only for being the critical parameter to obtain the Bond work index but also for characterizing the ore breakage behavior.

## 4. Conclusions

The following conclusions summarize the results obtained in this research:

- According to the obtained results, BBM and BRM grindability tests showed no continuity or clear correlation when considering $w_i$ values versus $P_{100}$, but a clear tendency was obtained in all cases when plotting *gpr* versus $P_{100}$.
- It is advised that energy consumption modelling based on correlations involving $w_i$ and other mechanical or operational parameters would yield a better determination coefficient using *gpr* values instead.
- The re-signifying of *gpr* evidenced to characterize the ore breakage behavior and its origin justify the proposal of naming *gpr* as the Maxson grindability index.

**Supplementary Materials:** The following are available online at www.mdpi.com/2075-4701/11/6/970/s1, Table S1. PENOUTA (tailings pond) BRM, Table S2. PENOUTA (tailings pond) BBM, Table S3. PENOUTA (mine) BRM, Table S4. PENOUTA (mine) BBM, Table S5. MITTERSILL BRM, Table S6. MITTERSILL BBM, Table S7. PANASQUEIRA BRM, Table S8. PANASQUEIRA BBM, Table S9. BARRUECOPARDO BRM, Table S10. BARRUECOPARDO BBM (test A), Table S11. BARRUECOPARDO BBM (test B).

**Author Contributions:** Conceptualization, A.L.C.-V. and J.M.M.-A.; methodology, G.G.G. and J.M.M.-A.; validation, G.G.G. and J.M.M.-A.; formal analysis, G.G.G., E.G. and H.A.; investigation, G.G.G., E.G. and H.A.; resources, J.O. and J.M.M.-A.; writing—original draft preparation, G.G.G. and J.M.M.-A.; writing—review and editing, A.L.C.-V., J.O. and J.M.M.-A.; visualization, G.G.G. and J.M.M.-A.; supervision, A.L.C.-V.; project administration, J.O. and J.M.M.-A.; funding acquisition, J.O. and J.M.M.-A. All authors have read and agreed to the published version of the manuscript.

**Funding:** This work is part of the OptimOre project funded by the European Union Horizon 2020 Research and Innovation Programme under grant agreement No 642201.

**Institutional Review Board Statement:** Not applicable.

**Informed Consent Statement:** Not applicable.

**Data Availability Statement:** Not applicable.

**Conflicts of Interest:** The authors declare no conflict of interest.

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
