# Peer review of "Variability Study of Bond Work Index and Grindability Index on Various Critical Metal Ores"

_metals, doi:10.3390/met11060970_

Round 1

Reviewer 1 Report

The authors compared the relationship between two indices, bond work and grindability, and the grinding size of different critical metal ores. Results show that the grindability index is more relevant than the bond work index. This work can provide guidance for the study of the correlation between ore grindability and rock mechanical parameters. However, I think it still has some deficiencies and I recommend a revision before acceptable publication. Detailed comments are listed below:

  1. What is the essential difference and advantage between BRM and BBM in the variability analysis? Please clarify.
  2. There are too few points in the figure for correlation analysis. Are the results convincing enough?
  3. The figures for analyzing the data are of one type and the key elements are consistent. It recommended merging some figures to save space and facilitate comparison.
  4. Regarding the materials and experimental data, they should be given by a table as well. This information is missing now.
  5. Are there any other indices that correlate with grinding size?
  6. The abstract of the manuscripts is very clear. However, the conclusion is not in-depth enough and needs to be revised.
  7. The fonts and formatting of the formulas and figures in the text are different. Please unify them.
  8. The English expression in some places is inappropriate and complicated. Such as “In order to equalise segregation at the mill ends, it is rotated level for eight revolutions, then tilted up 5º for one revolution, down 5º for another revolution and returned to level for eight revolutions continuously through each grinding cycle” It is noted that the manuscript needs careful editing by someone with expertise in technical English editing paying particular.

Author Response

R: The authors compared the relationship between two indices, bond work and grindability, and the grinding size of different critical metal ores. Results show that the grindability index is more relevant than the bond work index. This work can provide guidance for the study of the correlation between ore grindability and rock mechanical parameters. However, I think it still has some deficiencies and I recommend a revision before acceptable publication. Detailed comments are listed below:

R: What is the essential difference and advantage between BRM and BBM in the variability analysis? Please clarify.

A: Thank you to the reviewer for the comments. The authors consider that the variability study (both BRM and BBM work index calculation) is very important because, in practical applications of Bond’s methodology, sometimes the wi is considered as a constant value. Results show clearly that ore grindability varies, so in order to evidence the variation at a wide size range were considered both Bond standard tests (BBM for fine grinding and BRM for coarse grinding), and studied the correlation between wi or gpr with P100.

R: There are too few points in the figure for correlation analysis. Are the results convincing enough?

A: Considering that usually it is wrongly considered the work index as a constant, two or more points are needed to show that conceptual mistake. Each point was obtained from a full Bond grindability test (BRM or BBM), and when analysing all points globally, a minimum of 4-5 points are considered, which is enough to establish a correlation analysis.

R: The figures for analysing the data are of one type and the key elements are consistent. It recommended merging some figures to save space and facilitate comparison.

A: Thank you for the comment. Authors considered also merging figures (provided that they are also included in the Excel file – supplementary materials), but the results obtained in the different cases were so different when plotting wi versus P100, and so similar when plotting gpr versus P100 that finally it was decided to remain like they are.

R: Regarding the materials and experimental data, they should be given by a table as well. This information is missing now.

A: Thank you for the comment. Perhaps the reviewer does not realise that, along with the manuscript, it was provided an Excel spreadsheet as supplementary materials which will be available for readers. Authors consider that full information details are available considering the supplementary materials provided.

R: Are there any other indices that correlate with grinding size?

A: The Bond work index is the most practical and widely used. There are other methodologies and indices, but only known at an academic level and always their validation is performed against the Bond work index.

R: The abstract of the manuscripts is very clear. However, the conclusion is not in-depth enough and needs to be revised.

A: Thank you for the comment. According to the Metals instructions to authors, it is not mandatory to include a conclusions section, providing that a sound discussion is included in the Results section. In this case, authors consider that the considerations included in the Results and Discussion section are well justified and the discussion shows clearly the implications of the obtained results. The conclusion section has been included just as a summary of the discussion presented. However, some changes have been made in the revised version of the manuscript to improve this section

R:The fonts and formatting of the formulas and figures in the text are different. Please unify them.

A: Thank you for the suggestion. The fonts are selected automatically by Microsoft Equation editor and Microsoft Excel; they will be adapted (if finally accepted) in the final formatting process.

R:The English expression in some places is inappropriate and complicated. Such as “In order to equalise segregation at the mill ends, it is rotated level for eight revolutions, then tilted up 5º for one revolution, down 5º for another revolution and returned to level for eight revolutions continuously through each grinding cycle” It is noted that the manuscript needs careful editing by someone with expertise in technical English editing paying particular.

A: Thank you for the comment. The full paper was revised again by an English native technical translator and marked in yellow the corrections. However, the phrase pointed by the reviewer has been reproduced literally from the Bond’s original paper, and authors prefer to keep it the same.

Reviewer 2 Report

Authors have presented a compact experimental study of Bond grindability tests with selected ores

In Figure 7 both x- and y-axis titles must be written in English

Figures 7, 10 13 an 16: Straight connection between Rod Mill and Ball mill test results (gpr as a function of P100) can not be done, only simulation function can be continuous for whole range.

Row 278: in brackets is written wi, which should be wi

Authors could introduce more detailed all three specific energy consumption models (Rittinger, Kick and Bond), especially the size rangers, each one is valid and how they differ in practice from each other

Author Response

R: Authors have presented a compact experimental study of Bond grindability tests with selected ores

R: In Figure 7 both x- and y-axis titles must be written in English

A: Thank you very much, sorry for the translation mistake.

R: Figures 7, 10 13 an 16: Straight connection between Rod Mill and Ball mill test results (gpr as a function of P100) can not be done, only simulation function can be continuous for whole range.

A: Thank you for your comment. What we are trying to show is that, despite BRM and BBM tests are different and have different validity ranges, the gpr value obtained can be adjusted to a function, even when the analysis of wi with P100 does not show a clear correlation.

R: Row 278: in brackets is written wi, which should be wi

A: Thank you for the comment, it has been corrected in the revised version of the manuscript

R: Authors could introduce more detailed all three specific energy consumption models (Rittinger, Kick and Bond), especially the size rangers, each one is valid and how they differ in practice from each other

A: Thank you for the suggestion. It has been included, in the revised version, the following: “Subsequent studies (Hukki, 1961) explained that the three laws can be derived from a generalized comminution differential equation, having each one its better application size range (Rittinger’s law for fine grinding; Bond’s law for coarse grinding and secondary/tertiary crushing; and Kick’s law for primary crushing).

Round 2

Reviewer 1 Report

In the previous review, several suggestions were made to enable the authors to enhance the quality of the paper. The authors have diligently responded to my suggestions and made some changes to the content. The quality of this manuscript has been improved.

For the third comment in the first round of review, the author gives a reply and explanation. However, I think the figures are still too simple and lack of ornamental. Even if the figures can't be merged, it is recommended that the authors display them as subgraphs. That would reduce the number of figures and the quality of each graph will also be higher.

Author Response

R: In the previous review, several suggestions were made to enable the authors to enhance the quality of the paper. The authors have diligently responded to my suggestions and made some changes to the content. The quality of this manuscript has been improved.

For the third comment in the first round of review, the author gives a reply and explanation. However, I think the figures are still too simple and lack of ornamental. Even if the figures can't be merged, it is recommended that the authors display them as subgraphs. That would reduce the number of figures and the quality of each graph will also be higher.

A: Authors appreciate the reviewer comments. In the newly revised version of the manuscript, figures depicting gbp versus particle size were merged for each ore. We agree that the quality was improved following the reviewer suggestion.